# Curcumin Improves Neurogenesis in Alzheimer’s Disease Mice via the Upregulation of Wnt/β-Catenin and BDNF

**DOI:** 10.3390/ijms25105123

**Published:** 2024-05-08

**Authors:** Shengchun Lou, Danfeng Gong, Mengting Yang, Qing Qiu, Jialie Luo, Tingting Chen

**Affiliations:** Department of Pharmacology, School of Pharmacy, Nantong University, Nantong 226001, China; lawyer3396@stmail.ntu.edu.cn (S.L.); df20240406@163.com (D.G.); 18761725446@163.com (M.Y.); 19827031846@163.com (Q.Q.)

**Keywords:** curcumin, Alzheimer’s disease, adult neurogenesis, Wnt/β-catenin, BDNF, PI3K/Akt

## Abstract

Adult neurogenesis in the dentate gyrus (DG) is impaired during Alzheimer’s disease (AD) progression. Curcumin has been reported to reduce cell apoptosis and stimulate neurogenesis. This study aimed to investigate the influence of curcumin on adult neurogenesis in AD mice and its potential mechanism. Two-month-old male C57BL/6J mice were injected with soluble β-amyloid (Aβ_1–42_) using lateral ventricle stereolocalization to establish AD models. An immunofluorescence assay, including bromodeoxyuridine (BrdU), doublecortin (DCX), and neuron-specific nuclear antigen (NeuN), was used to detect hippocampal neurogenesis. Western blot and an enzyme-linked immunosorbent assay (ELISA) were used to test the expression of related proteins and the secretion of brain-derived neurotrophic factor (BDNF). A Morris water maze was used to detect the cognitive function of the mice. Our results showed that curcumin administration (100 mg/kg) rescued the impaired neurogenesis of Aβ_1–42_ mice, shown as enhanced BrdU^+^/DCX^+^ and BrdU^+^/NeuN^+^ cells in DG. In addition, curcumin regulated the phosphatidylinositol 3 kinase (PI3K)/protein kinase B (Akt) -mediated glycogen synthase kinase-3β (GSK3β) /Wingless/Integrated (Wnt)/β-catenin pathway and cyclic adenosine monophosphate response element-binding protein (CREB)/BDNF in Aβ_1–42_ mice. Inhibiting Wnt/β-catenin and depriving BDNF could reverse both the upregulated neurogenesis and cognitive function of curcumin-treated Aβ_1–42_ mice. In conclusion, our study indicates that curcumin, through targeting PI3K/Akt, regulates GSK3β/Wnt/β-catenin and CREB/BDNF pathways, improving the adult neurogenesis of AD mice.

## 1. Introduction

Alzheimer’s disease (AD) is one of the most common neurodegenerative diseases, and the progressive loss of memory and cognitive impairment are its typical characteristics. The entorhinal cortex and hippocampus have a great role in AD etiology [1]; among them, the hippocampus is a key component for learning and memory [2]. The dentate gyrus (DG) is a hippocampal subfield that mediates neurogenesis, and its function declines with age [3]. In the DG of the adult hippocampus, stem cells can produce new neurons continuously. In this way, the DG is also called a “neurogenic niche”, and continuous regeneration is termed “adult hippocampal neurogenesis” (AHN) [4]. Importantly, studies have also found that AHN is impaired in the process of aging and neurodegenerative disorders; for example, in clinical or animal AD studies, deficits in AHN have been observed [5], which can induce impairments in cognition. Therefore, the exploration of the fundamental and clinical aspects of AHN provides substantial value.

The Wingless/Integrated (Wnt)/β-catenin cascade [6] is known for its critical influence in tissue repairment and stem cell regeneration activity in diverse organs. Its beneficial role renders Wnt/β-catenin a widely studied element in the pathogenesis and therapy of aging disease [7]. Wnt/β-catenin plays fundamental roles in different phases of AHN, including stem cell activation, migration, differentiation, etc. [8]. Previous studies have revealed that inhibiting Wnt in vivo virtually blocks adult hippocampal neurogenesis; conversely, the overexpression of Wnt3a improves AHN in vivo and in vitro [9]. The deletion of a selective Wnt inhibitor, frizzled-related protein 3 (sFRP3), facilitates the maturation and synaptic morphology formation of neonatal neurons in the hippocampus of adult mice [10]. The Wnt pathway mediates gene transcription through the accumulation and transport of β-catenin to the nucleus. Previous studies have reported that enhanced glycogen synthase kinase-3β (GSK3β) phosphorylation stabilizes β-catenin and translocates it to the nucleus, subsequently activating Wnt signaling [11].

The brain-derived neurotrophic factor (BDNF) is the most common and important neurotrophic factor in the hippocampus [12]. The proliferation, survival, and synapse morphology of neurons are all deeply mediated by BDNF, which makes it one of the most crucial factors in AHN [13]. Exercise or antidepressant treatment can enhance the content of BDNF, leading to the improvement of AHN [14]. In AD mice, β-amyloid (Aβ), one of the typical pathological features of this disease, can decrease the BDNF content in the hippocampus [15]; this occurs mostly by lowering phosphorylated cyclic adenosine monophosphate (cAMP) response element-binding protein (CREB) levels, downstream of and regulated via phosphatidylinositol 3 kinase (PI3K)/protein kinase B (Akt) pathways [16]. In fact, it has been observed that the brain or serum levels of BDNF appear to be beneficial markers for cognitive conditions [17].

Curcumin is a traditional bioactive substance from Asia; it is commonly known as turmeric [18] and is widely used in Asian cuisine. It has previously been widely recognized for its anti-inflammatory, antimicrobial, and wound-healing activities [19]. Due to its relatively low or null cytotoxicity, in recent years, some scholars have studied its biological activity in other diseases and found that it has good anti-tumor [20] and neuroprotective ability [21]. Studies have shown that curcumin reduces cell apoptosis and stimulates neurogenesis by downregulating *caspase-3* mRNA [22]. In addition, the significant biological function and neuroprotective effects of curcumin in Parkinson’s disease occur via the activation of BDNF and PI3k/Akt signaling pathways [23]. Curcumin’s in vitro neuroprotective effects are mediated via the p62/keap-1/NF-E2-related factor 2 (Nrf2) and PI3K/Akt signaling pathway and autophagy inhibition [24]. In the pathological process of AD, whether curcumin can improve nerve regeneration and its related mechanisms is still unclear. Interestingly, phosphorylated Akt has been reported to inhibit the activity of GSK3β [25] and activate CREB to regulate BDNF transcription in the hippocampus [16]. We assume that curcumin has a fundamental role in neurogenesis via the targeting of the PI3K/Akt pathway.

This study will focus on exploring whether curcumin can improve AHN by regulating Wnt/β-catenin and BDNF in AD model mice, and it will verify the key factors and pathways in AHN.

## 2. Results

### 2.1. Decreased BrdU^+^ and BrdU^+^/DCX^+^ Cells in the DG of Aβ_1–42_ Mice Are Reversed via Curcumin Treatment

At 14 days after BrdU administration, the number of BrdU^+^ and BrdU^+^/DCX^+^ cells could be used to detect the number of neonatal neurons and the number of neonatal young neurons. At 28 days after BrdU administration, the numbers of BrdU+ cells and BrdU^+^/NeuN^+^ cells were measured to determine the number of new neurons in the mice that developed and matured (Figure 1A).

Confocal laser microscopy carried out on immunolabeled sections showed that, 14 days after BrdU administration, the BrdU^+^ cells in DG were significantly decreased in Aβ_1–42_ mice compared with control mice, which were treated via vehicles (F (1,28) = 10.94, *p* = 0.0026, Aβ_1–42_ vs. control: *p* < 0.0001), and this deficit in neurogenesis was reversed by curcumin (F (1,28) = 11.56, *p* = 0.0020, Aβ_1–42_ vs. Aβ_1–42_ +curcumin: *p* < 0.0001; Figure 1B). In addition, the number of BrdU^+^/DCX^+^ cells in the DG in Aβ_1–42_ mice was obviously reduced (F (1,28) = 7.881, *p* = 0.0090, Aβ_1–42_ vs. control: *p* = 0.0007), and this was reversed by curcumin (F (1,28) = 4.315, *p* = 0.0471, Aβ_1–42_ vs. Aβ_1–42_ +curcumin: *p* = 0.0028; Figure 1C). The immunofluorescence results after 14 days of BrdU administration revealed that the proliferation of neural progenitor cells and their differentiation into neurons was impaired in the DG of Aβ_1–42_ mice, and curcumin treatment improved it.

### 2.2. Decreased BrdU^+^ and BrdU^+^/NeuN^+^ Cells in the DG of Aβ_1–42_ Mice Are Reversed by Curcumin Treatment

Twenty-eight days after BrdU administration, the number of BrdU^+^ cells in the DG reflected the survival of newborn neurons. We observed that the number of BrdU^+^ cells was significantly decreased by almost 40% in Aβ_1–42_ mice compared to the control mice treated via vehicles (F (1,28) = 4.503, *p* = 0.0428, Aβ_1–42_ vs. control: *p* = 0.0001), and this was upregulated by curcumin (F (1,28) = 7.961, *p* = 0.0087, Aβ_1–42_ vs. Aβ_1–42_ +curcumin: *p* = 0.0005; Figure 2A). Similarly, the number of BrdU^+^/NeuN^+^ (neuron-specific nuclear antigen) cells in Aβ_1–42_ mice was reduced compared to that in control ones treated by vehicles (F (1,28) = 11.67, *p* = 0.0020, Aβ_1–42_ vs. control: *p* = 0.0023), which was improved by curcumin (F (1,28) = 10.63, *p* = 0.0029, Aβ_1–42_ vs. Aβ_1–42_ +curcumin: *p* = 0.0030; Figure 2B).

### 2.3. Curcumin Upregulates Wnt/β-Catenin Signaling and BDNF Content

Next, we further explore the underlying mechanisms by which curcumin improves adult hippocampus neurogenesis.

We examined the two most relevant factors, Wnt/β-catenin and BDNF. The stabilization and translocation of β-catenin were characterized by a reduction in phosphorylation and an increase in the total expression of β-catenin [11]. As shown in Figure 3A, β-catenin phosphorylation was increased in Aβ_1–42_ mice compared to the control mice treated via vehicles (F (1,28) = 5.521, *p* = 0.0261, Aβ_1–42_ vs. control: *p* = 0.0004), and this was reduced by curcumin (F (1,28) = 6.309, *p* = 0.0181, Aβ_1–42_ vs. Aβ_1–42_ +curcumin: *p* = 0.0003; Figure 3A). Conversely, the expression of total β-catenin was decreased in Aβ_1–42_ mice compared with control mice treated via vehicles (F (1,28) = 10.41, *p* = 0.0032, Aβ_1–42_ vs. control: *p* < 0.0001), and this was decreased by curcumin (F (1,28) = 12.83, *p* = 0.0013, Aβ_1–42_ vs. Aβ_1–42_ +curcumin: *p* < 0.0001; Figure 3B). These results show an increase in β-catenin activity due to curcumin in Aβ_1–42_ mice.

We also found that GSK3β phosphorylation was reduced in Aβ_1–42_ groups, indicating the enhanced activity of GSK3β (F (1,28) = 20.49, *p* = 0.0001, Aβ_1–42_ vs. control: *p* < 0.0001), which was increased by curcumin (F (1,28) = 15.03, *p* = 0.0001, Aβ_1–42_ vs. Aβ_1–42_ +curcumin: *p* < 0.0001; Figure 3C). Conclusively, curcumin could upregulate the Wnt/β-catenin pathway by reducing the activity of GSK3β in Aβ_1–42_ mice.

The ELISA study indicated that the BDNF content was also decreased in Aβ_1–42_ groups (F (1,28) = 11.12, *p* = 0.0024, Aβ_1–42_ vs. control: *p* = 0.0002), which was upregulated by curcumin (F (1,28) = 14.78, *p* = 0.0006, Aβ_1–42_ vs. Aβ_1–42_ +curcumin: *p* < 0.0001; Figure 3D). Moreover, CREB phosphorylation was decreased in Aβ_1–42_ groups compared with control mice (F (1,28) = 8.062, *p* = 0.0083, Aβ_1–42_ vs. control: *p* = 0.0005), and this was upregulated by curcumin (F (1,28) = 14.64, *p* = 0.0007, Aβ_1–42 vs_. Aβ_1–42_ +curcumin: *p* < 0.0001; Figure 3E). This part indicated that curcumin could upregulate BDNF content by increasing the activity of CREB.

### 2.4. Curcumin Upregulated Wnt/β-Catenin and BDNF Aβ_1–42_ Mice through the PI3K/Akt Pathway

Considering the close correlation between Wnt/β-catenin, BDNF, and PI3K/Akt [16,25], we assume that curcumin upregulates Wnt/β-catenin and BDNF through the PI3K/Akt pathway. Our study showed that Akt phosphorylation was decreased in Aβ_1–42_ mice (F (1,28) = 17.11, *p* = 0.0003, Aβ_1–42_ vs. control: *p* < 0.0001), which was reversed by curcumin (F (1,28) = 15.30, *p* = 0.0005, Aβ_1–42_ vs. Aβ_1–42_ +curcumin: *p* = 0.0001; Figure 4A). We further verified the role of Akt in curcumin-upregulated pathways. The results showed that curcumin enhanced the phosphorylation of GSK3β (F (1,28) = 4.524, *p* = 0.0424, Aβ_1–42_ vs. Aβ_1–42_ +curcumin: *p* = 0.0097; Figure 4B); reduced the phosphorylation of β-catenin (F (1,28) = 20.12, *p* = 0.0001, Aβ_1–42_ vs. Aβ_1–42_ +curcumin: *p* < 0.0001; Figure 4C); and enhanced the expression of β-catenin in Aβ_1–42_ mice (F (1,28) = 7.147, *p* = 0.0124, Aβ_1–42_ vs. Aβ_1–42_ +curcumin: *p* < 0.0011; Figure 4D). Moreover, these effects were reversed by the inhibitor of Akt: Ly294002 (phosphorylation of GSK3β: F (1,28) = 6.543, *p* = 0.0162, Aβ_1–42_ +curcumin vs. Aβ_1–42_ +curcumin+ Ly294002: *p* = 0.0045; phosphorylation of β-catenin: F (1,28) = 11.24, *p* = 0.0023, Aβ_1–42_ +curcumin vs. Aβ_1–42_ +curcumin+ Ly294002: *p* = 0.0003; β-catenin: F (1,28) = 10.48, *p* = 0.0031, Aβ_1–42_ +curcumin vs. Aβ_1–42_ +curcumin+ Ly294002: *p* = 0.0004). Similarly, curcumin enhanced the phosphorylation of CREB (F (1,28) = 4.277, *p* = 0.048, Aβ_1–42_ vs. Aβ_1–42_ +curcumin: *p* = 0.019; Figure 4E) and BDNF contents (F (1,28) = 14.13, *p* = 0.0008, Aβ_1–42_ vs. Aβ_1–42_ +curcumin: *p* < 0.0001; Figure 4F) in Aβ_1–42_ mice, and they were also both reversed by Ly294002 (CREB: F (1,28) = 5.032, *p* = 0.033, Aβ_1–42_ +curcumin vs. Aβ_1–42_ +curcumin+ Ly294002: *p* = 0.0141; BDNF: (F (1,28) = 23.52, *p* < 0.0001, Aβ_1–42_ +curcumin vs. Aβ_1–42_ +curcumin+ Ly294002: *p* < 0.0001). Above all, through different pathways, PI3K/Akt is the key target in the role of curcumin-enhanced Wnt/β-catenin and BDNF.

### 2.5. Curcumin-Upregulated Neurogenesis in Aβ_1–42_ Mice via Enhancing β-Catenin and BDNF

Finally, we verified the role of β-catenin and BDNF in curcumin-enhanced neurogenesis in Aβ_1–42_ mice. As described in the Materials and Methods section, the mice were pretreated by MSAB or Fc. As shown in Figure 5, MSAB or Fc pretreatment did not affect the amount of BrdU^+^/DCX^+^ cells in Aβ_1–42_ mice (F (2,42) = 18.06, *p* < 0.0001, Aβ_1–42_ vs. Aβ_1–42_ + MSAB: *p* = 0.9997; Aβ_1–42_ vs. Aβ_1–42_ + Fc: *p* = 0.9996; Figure 5A,C); in contrast, it reduced the expression of BrdU^+^/DCX^+^ cells in Aβ_1–42_ mice, which was administered via curcumin (Aβ_1–42_ +curcumin vs. Aβ_1–42_ +curcumin+ MSAB: *p* < 0.0001; Aβ_1–42_ +curcumin vs. Aβ_1–42_ +curcumin+ Fc: *p* < 0.0001; Figure 5A,C). Similarly, MSAB and Fc did not influence the amount of BrdU^+^/NeuN^+^ cells in Aβ_1–42_ mice (F (2,42) = 5.890, *p* = 0.0056, Aβ_1–42_ vs. Aβ_1–42_ + MSAB: *p* = 0.9597; Aβ_1–42_ vs. Aβ_1–42_ + Fc: *p* = 0.9971; Figure 5B,D); in contrast, the expression of BrdU^+^/NeuN^+^ cells in Aβ_1–42_ mice, which was administered via curcumin (Aβ_1–42_ +curcumin vs. Aβ_1–42_ +curcumin+ MSAB: *p* = 0.0009; Aβ_1–42_ +curcumin vs. Aβ_1–42_ +curcumin+ Fc: *=* 0.0001; Figure 5B,D), was reduced.

### 2.6. Curcumin Can Improve the Cognitive Function of Aβ_1–42_ Mice by Enhancing Wnt/β-Catenin and BDNF

With respect to the cognitive behavior of rodents, the Morris water maze tests spatial memory function, which is an important standard for reflecting learning and memory. To verify whether curcumin could improve the cognitive function of Aβ_1–42_ mice by enhancing Wnt/β-catenin and BDNF, we then performed the Morris water maze test. In the Morris water maze test, the staging latency of mice searching for visible platforms (1–2 test days) reflects search behavior or visual sensitivity, and the staging latency of mice who are searching for hidden platforms (3–7 test days) is used to judge spatial learning and memory function. A cruise exploration experiment was conducted 24 h (day 8) after the concealed platform test, and the swimming times of the mice were measured in four quadrants (the platform, its opposite quadrant, and the right and left adjacent quadrants) to estimate memory trace strengths, especially the residence time in the platform quadrant.

As shown in Figure 6A, the latency with respect to control and Aβ_1–42_ mice in terms of finding the visible platform was comparable (F (1,14) = 0.09016, *p* = 0.9926). There was no significant difference in the swimming speed between the two groups during training (F (1,14) = 0.08722, *p* = 0.9115; Bottom of Figure 6A). Compared with control mice, the spatial cognitive function of Aβ_1–42_ mice was significantly damaged, significantly prolonging the time required to find the hidden platform (F (1,14) = 28.8, *p* < 0.0001, day 4: *p* = 0.0024, day 5: *p* = 0.0003, day 6: *p* < 0.0001, day 7: *p* = 0.0005; Figure 6A).

As shown in Figure 6B, curcumin administration significantly shortened the time required to find the hidden platform in Aβ_1–42_ mice (F (5,42) = 6.870, *p* < 0.0001, Aβ_1–42_ vs. Aβ_1–42_ +curcumin: *p* = 0.0159, day 4: *p* = 0.0336, day 5: *p* = 0.0053, day 6: *p* = 0.0045, day 7: *p* = 0.0045; Figure 6B), and there was no significant difference between Aβ_1–42_ mice with Aβ_1–42_ +Fc mice and Aβ_1–42_ +MSAB mice with respect to the latency of seeking hidden platforms (Aβ_1–42_ vs. Aβ_1–42_ +MSAB: *p* = 0.9994; Aβ_1–42_ vs. Aβ_1–42_ + Fc: *p* > 0.999; Figure 6B), showing that Fc and MSAB administration had no significant effects on spatial cognitive function in Aβ_1–42_ mice. However, Fc and MSAB prolonged the time needed to search for hidden platforms in Aβ_1–42_ mice which had been administered curcumin (Aβ_1–42_ +curcumin vs. Aβ_1–42_ +curcumin+ MSAB: *p* < 0.0001. day 4: *p* = 0.0169; day 5: *p* = 0.0089; day 6: *p* = 0.0025; day 7: *p* = 0.0104. Aβ_1–42_ +curcumin vs. Aβ_1–42_ +curcumin+ Fc: *p* < 0.0001; day 4: *p* = 0.0195; day 5: *p* = 0.0015; day 6: *p* = 0.0005; day 7: *p* = 0.0006; Figure 6B).

In the probe test on day 8, the platform was removed, and the residence times of mice in different quadrants were recorded. As shown in Figure 6C, compared with control mice, Aβ_1–42_ mice showed prolonged time spent in the target quadrant (*p* = 0.0206, t = 2.610; Figure 6C), exhibiting deficits in cognitive function. In addition, the results showed that MSAB or Fc pretreatment did not affect the time spent in the target quadrant for Aβ_1–42_ mice (F (2,42) = 5.080, *p* = 0.0106, Aβ_1–42_ vs. Aβ_1–42_ + MSAB: *p* = 0.9998; Aβ_1–42_ vs. Aβ_1–42_ + Fc: *p* > 0.9999; Figure 6D); in contrast, the time spent in the target quadrant was prolonged with respect to Aβ_1–42_ mice who were administered curcumin (Aβ_1–42_ vs. Aβ_1–42_ +curcumin: *p* = 0.0159; Aβ_1–42_ +curcumin vs. Aβ_1–42_ +curcumin+ MSAB: *p* = 0.0139; Aβ_1–42_ +curcumin vs. Aβ_1–42_ +curcumin+ Fc: *p* = 0.0046; Figure 6D).

## 3. Discussion

In this study, we first reported the beneficial role of curcumin in the adult hippocampus neurogenesis through both Wnt/β-catenin and BDNF in Aβ_1–42_ mice by targeting PI3K/Akt, providing a new target and direction for drug therapy of AD.

### 3.1. The Regulatory Role of Wnt/β-Catenin in Adult Neurogenesis

Our study found that curcumin could enhance BrdU^+^/DCX^+^ cells after 14 days of Brdu administration and increase BrdU^+^/NeuN^+^ cells after 28 days of Brdu administration in the hippocampus of Aβ_1–42_ mice, which presented a great effect on neurogenesis. BrdU is a synthetic thymidine analogue that binds to cellular DNA in place of thymidine nucleoside during the S phase of cell division. DNA synthesis, cell division, and apoptosis were detected. DCX is a microtubule-associated protein which is used to label young, immature neurons [26]. DCX is specifically expressed in newly generated healthy neurons, peaks in the second week after neonatal birth, and decreases as mature neurons emerge, which are labeled by NeuN in the fourth week [27]. In this manner, our results revealed an effect of curcumin on the number of newly generated neurons and mature neurons.

The prenatal development of the mammalian brain consists of four stages [28]: the first stage, the generation of precursor cells from stem cells; the second stage, the migration of precursor cells to the target brain region; the third stage, the differentiation of new neurons into mature synaptic active cells; and the fourth stage, the pruning of new neurons via apoptosis. Therefore, factors that influence brain migration and differentiation in mammals play an important role in adult neurogenesis.

New cells in the anterior part of the sub-ventricular zone migrate to the olfactory bulb via anterior migration, where they differentiate into interneurons, granulosa neurons, and periglomerular neurons. Previous studies have observed low levels of β-catenin expression in adult transgenic mice, and olfactory bulbs in adult transgenic animals do not increase in size with age relative to normal mice, suggesting that the normal migration behavior of these neurons may be impaired or misdirected [29]. The study demonstrated the significant role of β-catenin in the migration of precursor cells. Unfortunately, a detailed and robust study of the mechanisms under this provision has not yet been undertaken.

Neuronal differentiation is controlled by intrinsic and extrinsic regulatory fields [30]. In addition to Wnt ligands, which regulate this process, β-catenin also has an effect on neurogenesis. The activation of β-catenin leads to the proliferation of the pool of neural progenitor cells, leading to the expansion of the entire neural tube field [31]. Xanthoceraside (XAN) treatment enhances the expression of Wnt3a, phosphorylated β-catenin and induces the nuclear translocation of β-catenin in the hippocampus of APP/PS1 mice, in addition to inducing NSC proliferation and neuronal differentiation in APPswe/PS11E9 mice [32]. In addition, the biological trace element Ethosuximide (ETH), which inactivates GSK3β and increased β-catenin levels, enhancing proliferation and neuronal differentiation in the DG of a rat model of AD [33]. The study reported the vital role of Wnt/β-catenin in neuronal differentiation and neurogenesis; however, the detailed mechanisms still require more exploration.

In the present study, curcumin enhanced the total expression of β-catenin and decreased its phosphorylation. β-catenin is the central component of the Wnt/β-catenin signaling pathway, which not only transmits information in the cytoplasm, but also translocates relative to the nucleus-activating target gene’s transcription. β-catenin phosphorylation inhibition prevents its ubiquitination and degradation [34]. β-catenin accumulates in the cytoplasm and translocates into the nucleus, where it performs its function [35,36]. Enhanced total expression and decreased phosphorylation both indicate the increased nuclear transportation of β-catenin, which mainly mediates gene transcription and influences neurogenesis. In this manner, curcumin might improve adult neurogenesis in Aβ_1–42_ mice by activating Wnt/β-catenin. Moreover, curcumin improves adult neurogenesis, which is abolished by the inhibitor of Wnt/β-catenin.

Furthermore, we explored the mechanism of Wnt/β-catenin enhanced by curcumin. We found that curcumin could also reduce the activity of GSK3β (shown as increased phosphorylation), which has been reported to reversely regulate β-catenin translocation [37]. In our study, the change in GSK3β was consistent with the alteration of the expression and phosphorylation of β-catenin, showing that curcumin could reduce the activity of GSK3β and activate Wnt/β-catenin. We proposed that curcumin treatment in Aβ_1–42_ mice reduced the activity of GSK3β and then induced a decrease in β-catenin phosphorylation and an increase in β-catenin expression, which indicated the activation of Wnt/β-catenin, consequently enhancing neurogenesis in Aβ_1–42_ mice.

However, Wnt/β-catenin pathways also could be regulated by other mechanisms, such as c-Jun N-terminal kinase (JNK) [38]. Whether curcumin also aims at other targets to mediate this pathway requires more research.

### 3.2. The Vital Role of PI3K/Akt in the Improvement of Neurogenesis via Curcumin Treatment in Aβ_1–42_ Mice

In this study, we found that PI3K/Akt is the key target and intermediate regulator. This is because curcumin, via the activation of PI3K/Akt, reduces the activity of GSK3β and then enhances the Wnt/β-catenin pathway on the one hand; on the other hand, it upregulates the activity of CREB and then increases the content of BDNF in Aβ_1–42_ mice. Both play a vital role in improving neurogenesis.

PI3K/Akt mediates multiple cellular functions, such as survival, proliferation, migration, and differentiation [27]. In the central nervous system (CNS), PI3K/Akt plays a fundamental role in regulating neuronal survival. In the axis, activated PI3K triggers Akt phosphorylation, resulting in CREB activation, which regulates BDNF directly, as described in the results [27]. Based on this result, we will further explore the effect of curcumin on neuron morphology in Aβ_1–42_ mice through Akt/CREB/BDNF because it may also affect the structure of neurons [15]. GSK3β, a kinase that is abundant in the brain, can promote neuronal apoptosis, and the dysregulation of GSK3β can destroy the development of neurons [39]. GSK3β is one of the substrates of Akt [40], which is inactivated when phosphorylated by the activated Akt at Ser9. In the present study, curcumin increased the phosphorylation of GSK3β via the activation of Akt and then regulated the Wnt//β-catenin pathway in Aβ_1–42_ mice. These findings revealed the critical role of PI3K/Akt in Aβ_1–42_ mice neurogenesis via the treatment of curcumin. However, the specific mechanism underlying the modulation of curcumin in PI3K/Akt remains unclear.

It has been reported that α7nAChR activation can directly increase the calcium influx [41] and regulate the PI3K/Akt and ERK pathways [42,43]. Our previous study also revealed the role of α7nAChR in regulating PI3K/Akt and BDNF content [15]. In that study, we observed that H-Ras inhibition could upregulate α7nAChR-dependent PI3K/Akt and Calmodulin-Dependent Protein Kinase II (CaMKII); however, Ras inhibition-enhanced BDNF content was related to CaMKII. The study also indicated the fundamental role of α7nAChR in hippocampal neurogenesis, since it could affect different pathways that mediate BDNF content in the hippocampus. A previous study also revealed the influence of curcumin on α7nAChR, and it reported the neuroprotective role of curcumin by mediating α7nAChR in Parkinson’s Disease [44,45], autistic-like social deficits [46], etc. [47,48]. We will further explore whether the curcumin-enhanced PI3K/Akt pathway is similarly related to α7nAChR modulation in Aβ_1–42_ mice.

Phosphatase and tensin homolog deleted on chromosome ten (PTEN) is a tumor suppressor gene and an inhibitor of PI3K/Akt which has been reported to modulate oncogenesis in the study of cancers [49]. PTEN can regulate cell proliferation, differentiation, growth, and apoptosis [50]; moreover, it can inhibit the growth and invasion of tumor cells by inhibiting PI3K/Akt [51]. Reports focused on cancers revealed that curcumin could increase PTEN and p53 expression, further inhibiting the activation of PI3K/Akt [52]. We will further explore the regulation of PTEN in curcumin-treated Aβ_1–42_ mice and observe the role of PTEN in downstream Akt/GSK3β/β-catenin and Akt/CREB/BDNF.

Taken together, our findings indicate that curcumin can improve adult neurogenesis in Aβ_1–42_ mice by targeting PI3K/Akt, which in turn regulates GSK3β/β-catenin and CREB/BDNF. In this manner, cognitive function can be enhanced.

## 4. Materials and Methods

### 4.1. Experimental Animals

All experiments that were conducted on animals were approved by the Animal Experiment Ethics Committee of Nantong University, and the experimenters in the research group strictly implemented the welfare and ethics regulations regarding experimental animals during the animal experiments. In this study, 2-month-old C57BL/6J male mice (SLAC Laboratory Animal Co., Ltd., Shanghai, China) were used as experimental animals. The animals were kept in a stable and good environment in the Animal Center of Nantong University, which was conducive to the stability of various indicators. The temperature was maintained at 23 ± 2 °C. Relative humidity was maintained at 55 ± 5%. Moreover, the light/dark time cycle was as follows: 12:12 h. During the experiment, the mice were allowed to move freely in the cage, and food and water intake were provided ad libitum. The total number of mice was 288, weighing approximately 25 g.

### 4.2. AD Model Preparation

The specific process of AD model preparation was reported in our previously published article, and the process was mainly completed through the administration of Aβ_1–42_ (Sigma, St. Louis, MO, USA) via intracerebroventricular injection.

Briefly, Aβ_1–42_ was dissolved in 1,1,1,3,3,3-hexafluoro-2-propanol (HFIP, Sigma, St. Louis, MO, USA), flash-frozen in liquid nitrogen, and then lyophilized to completely remove the solvent. The lyophilized Aβ_1–42_ peptides were then dissolved in 100 mM NaOH at 6 mg/mL, aliquoted, flash-frozen in liquid nitrogen, and stored at −80 °C until use [53].

For the intracerebroventricular injection (i.c.v.) of soluble Aβ_1–42_, mice were intraperitoneally anesthetized with isoflurane and then placed into a stereotactic apparatus (Motorized Stereotaxic Stereo Drive; KOPF 900HD, Neurostar, Berlin, Germany). Freshly prepared Aβ_1–42_ (0.3 nmol/2 μL in 0.1 M phosphate-buffered saline [PBS]) was injected into bilateral cerebral ventricles (0.3 mm posterior, 1.0 mm lateral, and 2.5 mm ventral to the bregma) using a stepper-motorized microsyringe at 0.2 μL/min. The aggregated Aβ_1–42_ in the hippocampus was confirmed via immunostaining with an Aβ-specific antibody [54]. Control mice were treated in the same way and injected with the same volume of PBS.

### 4.3. Drug Administration

Curcumin (Sigma, St. Louis, MO, USA) was dissolved in 0.1% CMC-Na, and intragastric (i.g.) administration at 100 mg/kg [55] was carried out for 14 days.

LY294002 (inhibitor of PI3K/Akt) and MSAB (inhibitor of Wnt/β-catenin) were dissolved using dimethyl sulfoxide (0.5%), and they were administered 30 min before curcumin administration. Ly294002 (0.3 nmol/mouse) or MSAB (0.6 nmol/mouse) was administered in the right ventricle via repeated intracerebroventricular injections, in accordance with [15]. For repeated intracerebroventricular injections, a 28 G stainless-steel guide cannula (Plastic One) was implanted into the right lateral ventricle (0.3 mm posterior, 1.0 mm lateral, and 2.5 mm ventral to the bregma) and anchored to the skull with three stainless steel screws and dental cement [56]. Control mice were injected with the same volume of saline containing 0.5% dimethyl sulfoxide.

Recombinant mouse Trobe/Fc chimera protein (Fc), which can sequester endogenous BDNF, was purchased from R&D Systems (Minneapolis, MN, USA). It was dissolved in sterile PBS and 0.9% sterile physiological saline and injected intracerebroventricularly (10 ng/mouse [15]) immediately before Aβ injection once daily for 14 consecutive days, 30 min before curcumin administration.

### 4.4. Western Blotting

After the mice had been anesthetized with ether, the brain was severed, and the hippocampus was taken as a backup. The hippocampal tissues or brain slices were homogenized in a lysis buffer containing 50 mM Tris–HCl (pH 7.5), 150 mM NaCl, 5 mM EDTA, 10 mM NaF, 1 mM sodium orthovanadate, 1% Triton X-100, 0.5% sodium deoxycholate, 1 mM phenylmethylsulfonyl fluoride, and protease inhibitor cocktail (Complete; Roche, Mannheim, Germany), followed by incubation for 30 min at 4 °C. After sonication, the samples were centrifuged at 12,000 rpm for 15 min at 4 °C, and the supernatant was harvested. The protein concentration was determined using the BCA Protein Assay Kit (Pierce Biotechnology Inc., Rockford, IL, USA). Then, proteins of equal amounts were mixed with a loading buffer and boiled for 5 min.

Proteins of equal amounts (20 g) were separated via SDS–polyacrylamide gel electrophoresis (SDS–PAGE) and then transferred onto a polyvinylidene fluoride (PVDF) membrane, which was subsequently incubated with 5% nonfat milk for 60 min at room temperature. After washing was carried out three times, the membrane was incubated overnight at 4 °C with primary antibodies. GAPDH or β-actin served as the internal control. Appropriate horse radish peroxidase (HRP)-conjugated secondary antibody was used for detection via enhanced chemiluminescence (Pierce). ImageJ 1.52p (NIH Image, Bethesda, MD, USA) was used to determine protein expression, which was normalized to the expression of the internal control. The antibodies used are all listed in Appendix A.

### 4.5. Immunochemistry Examination

BrdU, doublecortin (DCX), and NeuN staining was carried out according to the steps mentioned in our previous article [27]. Three intraperitoneal injections (i.p.) of bromodeoxyuridine (BrdU, 50 mg/kg; Sigma Aldrich, St. Louis, MO, USA) were carried out at intervals of 8 h. The animals were anesthetized with isoflurane and transcardially perfused with 4% phosphate-buffered paraformaldehyde. Brain coronal sections (40 μm) were cut using a vibrating microtome. Every 5th free-floating section was treated with 2 M HCl at 37 °C for 30 min.

For BrdU immunostaining, the sections were treated with 3% normal donkey serum for 45 min and then incubated in a rat monoclonal anti-BrdU antibody at 4 °C overnight. After several PBS rinses, the sections were incubated in Alexa Fluor 594 AffiniPure Donkey Anti-Rat lgG(H+L) under agitation for 2 h. Immunoreactivities were visualized using the avidin–biotin–horseradish peroxidase complex (ABC Elite) and 3,3′-diaminobenzidine (DAB). For BrdU and NeuN or doublecortin (DCX) double immunostaining, the sections were incubated with a rat monoclonal anti-BrdU antibody, which was revealed using a CY3-labeled anti-rat IgG antibody; the rabbit monoclonal anti-Neun antibody, or rabbit polyclonal anti-DCX antibody, was revealed using Alexa Fluor 594 AffiniPure Donkey Anti-Rat lgG(H+L) and Alexa Fluor 488 AffiniPure Donkey Anti-Rabbit lgG(H+L). The sections were mounted onto subbed slides and stored at 4 °C.

In order to avoid any bias, specimen analyzers were blinded to the examination. (1) Immuno-positive BrdU (BrdU^+^) cells were observed at every 5th section (200 μm apart) using a conventional light microscope (Olympus DP70, Tokyo, Japan) with a 60× objective. The total number of BrdU^+^ cells per section was counted and multiplied by 5 to obtain the total number of cells in DG. (2) BrdU^+^/NeuN^+^ and BrdU^+^/DCX^+^ cells were assessed using a confocal laser-scanning microscope (Leica TCS SP8, Heidelberg, Germany).

The details of the antibodies and primary reagents are described in Appendix A.

### 4.6. Enzyme-Linked Immunosorbent Assay (ELISA)

The released BDNF content was detected via ELISA. The procedure was carried out according to the steps mentioned in our previous article [15]. The fresh mouse hippocampus tissues were placed into an ELISA special homogenate buffer that had been prepared in advance. The tissues were ground with a freeze grinder, and then they were centrifuged at 4 °C and 13,500 rpm for 15 min before absorbing the supernatant. Our goal was to detect the mature BDNF content rather than the total BDNF content. Thus, the samples were not treated with hydrochloric acid. The samples and BDNF standard were added to the well plate, and then the anti-BDNF specific antibody was added. Moreover, the samples were co-incubated in a shaker for 2 h. After incubation, the samples were washed, and the second antibodies labeled with HRP were added and incubated in the shaker for 1 h. Finally, the reaction substrate of peroxidase (yetramethylbenzidine provided by the kit) was dripped onto the pore plate. After 7 min of reaction, the terminating solution was added, and the liquid was gently oscillated to make it uniform. Absorbance was measured with an enzymoleter, and the specific value was calculated using the standard curve method.

### 4.7. Morris Water Maze

The MWM test was conducted for eight days consecutively, referring to the report published previously [15], to reflect the spatial cognitive function of mice [57].

A full MWM test typically lasts eight days, consisting of an initial two days of visible platform training, five days of hidden platform training, and a final day of spatial probe test. The MWM tests were carried out in a circular pool that was 120 cm in diameter, which was mounted with a video camera and connected to a computer equipped with a video tracking system to record and analyze the swimming tracks of the mice. The pool was divided into four quadrants and dyed white with white food coloring. The water in the pool needed to be kept stable, and the water temperature could be controlled at 23 ± 2 °C. A cylindrical, dark-colored platform (7 cm in diameter) was placed into one of the quadrants of the pool. During the first two days of the experiment, the platform was 0.5 cm above the water surface during visible platform training. The platform in the pool remained approximately 1 cm underwater during hidden platform training, and was removed from the pool during the spatial probe test on day 8.

The latency relative to reaching visible or hidden platforms and the swimming distance were measured. During training, each mouse was placed in one of the four quadrants randomly, with its head toward the wall. The video analysis system (XR-XM101, software purchased from Shanghai XinRuan Information Technology Co., LTD, Shanghai, China) installed above the pool began recording a video of the mice swimming as the experimenters released the mice, and the video was analyzed simultaneously. Mice could swim in the pool for a maximum of 90 s. If the mice could board the platform within the maximum swimming time, they were removed from the platform immediately, and the video recording stopped. If the mouse failed to find the platform within 90 s, it was guided to the platform, and the test was terminated. After removing the mice from the pool, the mice were properly disposed of to ensure that they were in a healthy state. These tests are conducted four times a day in different quadrants of the pool and at intervals of about 30 min.

The spatial probe test was performed on day 8. During the spatial probe test, the platform was removed from the pool, the mice were allowed to swim freely for 90 s from the furthest quadrant relative to the platform, and the swimming track was recorded. The recorded video was used to analyze the spatial reference memory retention of the mice, and the percentage of time consumed in each quadrant relative to the total swimming time was calculated.

### 4.8. Inclusion and Exclusion Criteria

The animals were included in the study if they underwent successful Aβ_1–42_ intracerebroventricular injection, defined by a great decline in cognition in the MWM test. The animals were excluded if the stainless-steel guide cannula became dislodged during the administration of inhibitors or Fc, or if the animal died prematurely, preventing the collection of behavioral and histological data.

### 4.9. Statistical Analysis

In this study, statistical and ANOVA analyses were carried out using GraphPad Prism 8, and the data used in the bar chart are presented as the mean value ± standard error. The *p*- and *F*-values are described in the Results section. Among them, *p* < 0.05 was considered statistically significant.

## 5. Conclusions

In conclusion, our study indicates that curcumin decreases the activity of GSK3β in Aβ_1–42_ mice by targeting PI3K/Akt, enhancing Wnt/β-catenin; moreover, it increases the activity of CREB, enhancing the BDNF content, which consequently improves adult neurogenesis and the spatial cognitive function of AD mice (Graphical Abstract). This study revealed the effects of curcumin on neurogenesis and cognitive behavior in AD mice, providing a sufficient theoretical basis for the application of curcumin in AD.

## Figures and Tables

**Figure 1 ijms-25-05123-f001:**
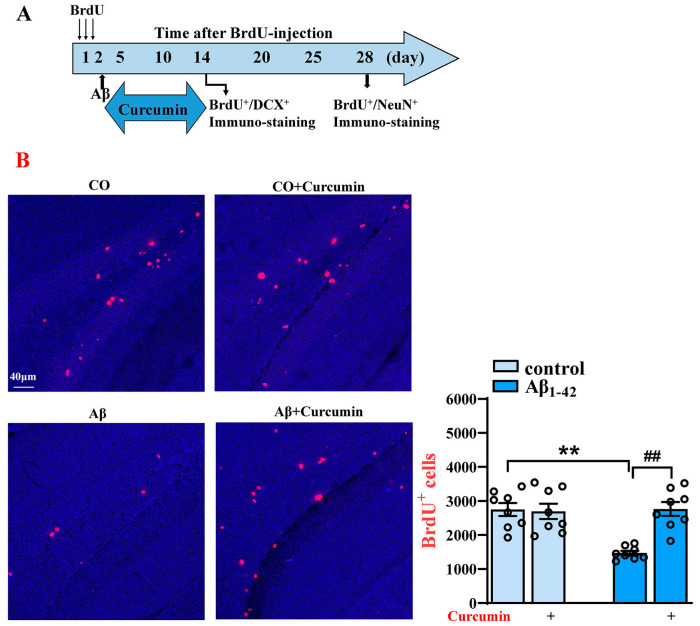
Decreased BrdU^+^ and BrdU^+^/DCX^+^ cell levels in the DG of Aβ_1–42_ mice after the administration of BrdU 14 days are reversed by the curcumin treatment. (**A**) Flowchart of the experiment; the time for the administration of BRDU, Aβ, and curcumin; and the time for the immuno-staining of BrdU^+^/DCX^+^ and BrdU^+^/NeuN^+^. (**B**) Immunostaining of BrdU^+^ cells (14 days after BrdU treatment) in different groups (right) and a columnar statistical chart. (**C**) Immunostaining of BrdU^+^, DCX^+^, and BrdU^+^/DCX^+^ cells (14 days after BrdU treatment) in different groups and columnar statistical chart (down). CO: control mice; red signals: BrdU^+^ cells; green signals: DCX^+^ cells; blue background: Dapi staining. The white arrows: cells marked by BrdU^+^/DCX^+^. (**B**,**C**) *n* = 8 mice per group, two-way Analysis of Variance (ANOVA), followed by Sidak’s test. ** *p* < 0.01 vs. control mice; ^##^
*p* < 0.01 vs. Aβ_1–42_ mice.

**Figure 2 ijms-25-05123-f002:**
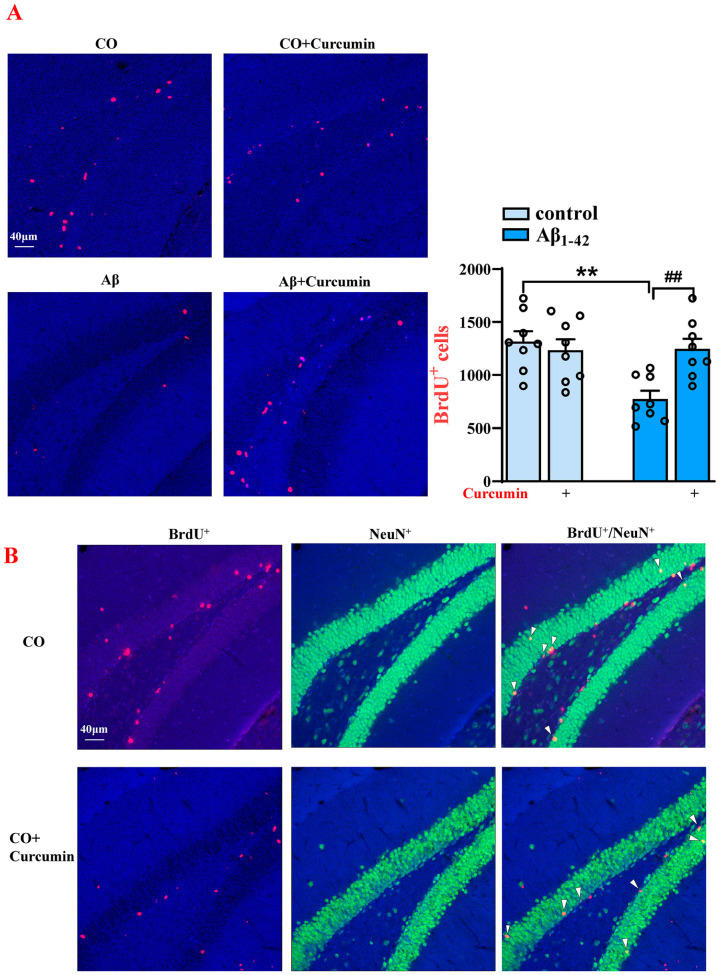
Decreased BrdU^+^ and BrdU^+^/NeuN^+^ cell levels in the DG of Aβ_1–42_ mice after the administration of BrdU for 28 days were reversed via the curcumin treatment. (**A**) Immunostaining of BrdU^+^ cells (28 days after BrdU treatment) in different groups (right) and a columnar statistical chart. (**B**) Immunostaining of BrdU^+^, NeuN^+^, and BrdU^+^/NeuN^+^ cells (28 days after BrdU treatment) in different groups and a columnar statistical chart (down). CO: control mice; red signals: BrdU^+^ cells; green signals: NeuN^+^ cells; blue background: Dapi staining. The white arrows: cells marked by BrdU^+^/ NeuN^+^. (**A**,**B**) *n* = 8 mice per group, two-way ANOVA, followed by Sidak’s test. ** *p* < 0.01 vs. control mice; ^##^
*p* < 0.01 vs. Aβ_1–42_ mice.

**Figure 3 ijms-25-05123-f003:**
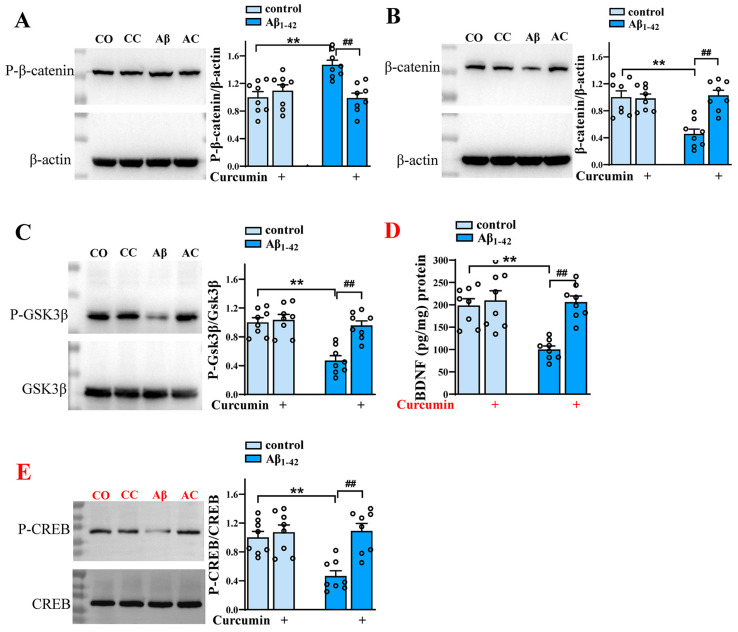
Curcumin upregulates Wnt/β-catenin signaling and BDNF content. The expression of hippocampal phospho-β-catenin (**A**), β-catenin (**B**), phospho-GSK3β (**C**), and phospho-CREB (**E**) in control mice and Aβ_1–42_ mice administered curcumin or vehicle. CO: control mice; CC: control +curcumin mice; Aβ: Aβ_1–42_-mice; AC: Aβ_1–42_-mice +curcumin. (**D**) The BDNF content (%) in the hippocampus in control mice and Aβ_1–42_-mice administered curcumin or vehicle. (**A**–**E**) *n* = 8 mice per group, two-way ANOVA, followed by Sidak’s test. ** *p* < 0.01 vs. control mice; ^##^
*p* < 0.01 vs. Aβ_1–42_ mice.

**Figure 4 ijms-25-05123-f004:**
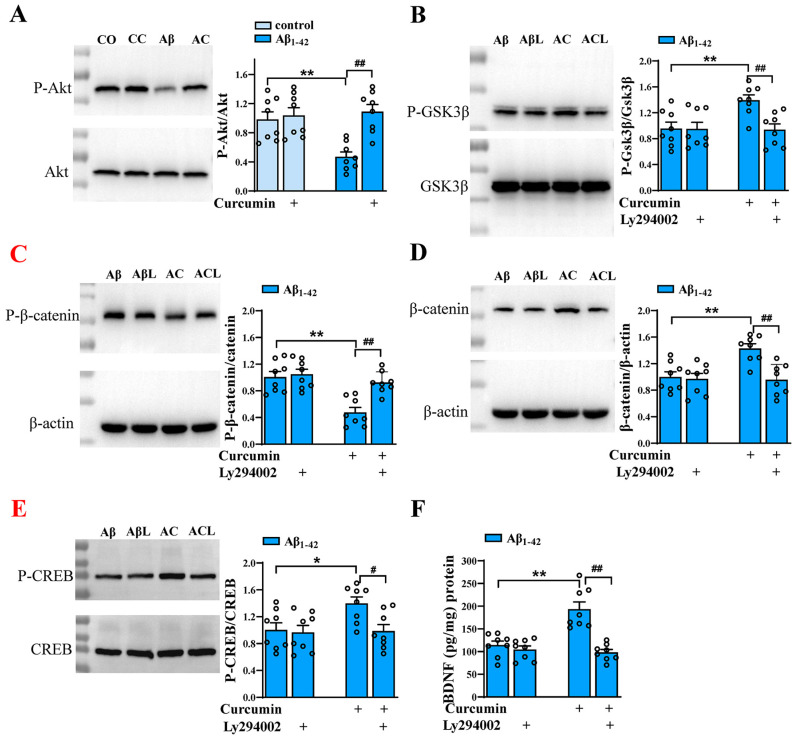
Curcumin upregulated Wnt/β-catenin and BDNF in Aβ_1–42_ mice through the PI3K/Akt pathway. (**A**) Expressions of hippocampal phospho-Akt in control mice and Aβ_1–42_ mice administered via curcumin or vehicle. Hippocampal phospho-GSK3β (**B**), phospho-β-catenin (**C**), and phospho-CREB (**D**) in Aβ_1–42_ mice treated with or without curcumin or Ly294002. CO: control mice; CC: control+ curcumin mice; Aβ: Aβ_1–42_-mice; AC: Aβ_1–42_-mice +curcumin. (**E**) The BDNF content (%) in the hippocampus in Aβ_1–42_ mice treated with or without curcumin or Ly294002. (**A**–**F**) *n* = 8 mice per group, two-way ANOVA, followed by Sidak’s test. (**A**) ** *p* < 0.01 vs. control mice; ^##^
*p* < 0.01 vs. Aβ_1–42_ mice; (**B**–**F**) * *p* < 0.05 or ** *p* < 0.01 vs. Aβ_1–42_ mice; ^#^
*p* < 0.05 or ^##^
*p* < 0.01 vs. Aβ_1–42_ mice +curcumin.

**Figure 5 ijms-25-05123-f005:**
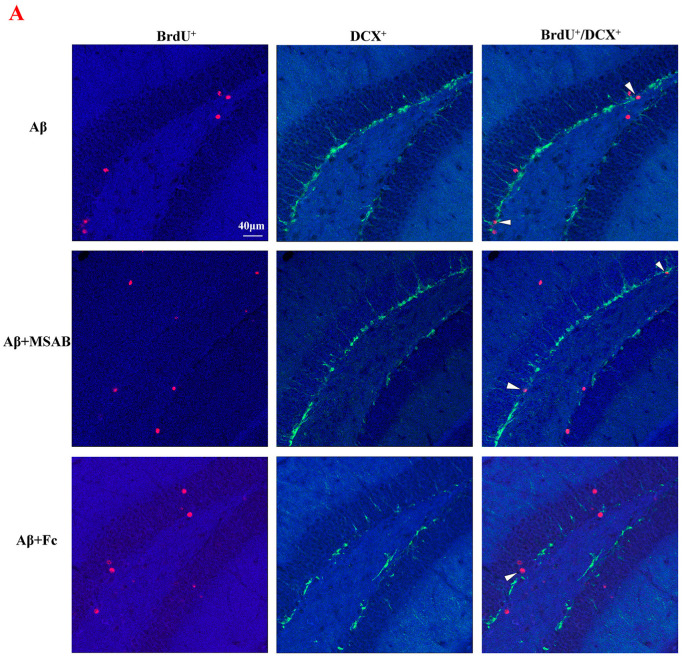
Curcumin-upregulated neurogenesis in Aβ_1–42_ mice via enhancing β-catenin and BDNF. (**A**) Immunostaining of BrdU^+^, DCX^+^, and BrdU^+^/DCX^+^ cells in Aβ_1–42_, Aβ_1–42_+MSAB, Aβ_1–42_+Fc, Aβ_1–42_ +curcumin, Aβ_1–42_+curcumin+MSAB, and Aβ_1–42_ +curcumin+Fc mice. Red signals: BrdU^+^ cells; green signals: DCX^+^ cells; blue background: Dapi staining. (**B**) Immunostaining of BrdU^+^, NeuN ^+^, and BrdU^+^/NeuN^+^ cells in Aβ_1–42_, Aβ_1–42_+MSAB, Aβ_1–42_+Fc, Aβ_1–42_ +curcumin, Aβ_1–42_ +curcumin+MSAB, and Aβ_1–42_ +curcumin+Fc mice. Red signals: BrdU^+^ cells; green signals: NeuN^+^ cells; blue background: Dapi staining. The white arrows: cells marked by BrdU^+^/ NeuN^+^. (**C**) Columnar statistical chart for (**A**). (**D**) Columnar statistical chart for (**B**). (**C**,**D**) *n* = 8 mice per group, two-way ANOVA, followed by Tukey’s test. ** *p* < 0.01 vs. Aβ_1–42_ mice; ^##^
*p* < 0.01 vs. Aβ_1–42_ mice+curcumin.

**Figure 6 ijms-25-05123-f006:**
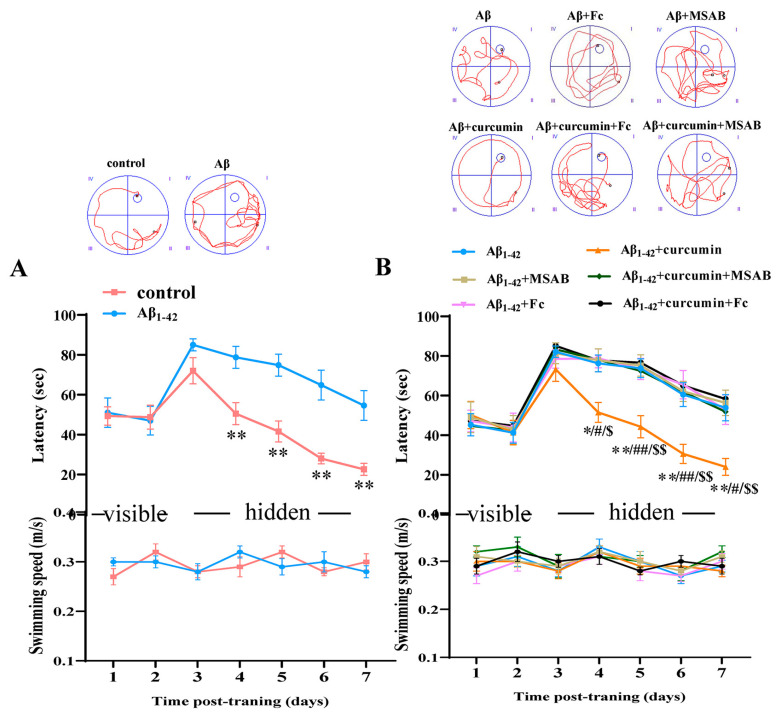
Curcumin upregulated cognitive function in Aβ_1–42_ mice via enhancing β-catenin and BDNF. (**A**) Latency in finding the visible and hidden platforms in the MWM test of the control and Aβ_1–42_ mice (upper). n = 8 mice per group, repeated measurement ANOVA, followed by Sidak’s test; ** *p* < 0.01 vs. control mice. Swimming speed of the control and Aβ_1–42_ mice (down). (**B**) Latency in finding the visible platform and hidden platform in the MWM test of Aβ_1–42_, Aβ_1–42_+MSAB, Aβ_1–42_+Fc, Aβ_1–42_ + curcumin, Aβ_1–42_ + curcumin + MSAB, and Aβ_1–42_ + curcumin + Fc mice (upper). *n* = 8 mice per group, repeated measure three-way ANOVA, followed by Tukey’s test. * *p* < 0.05; ** *p* < 0.01, Aβ_1–42_ vs. Aβ_1–42_ +curcumin mice. ^#^
*p* < 0.05; ^##^
*p* < 0.01, Aβ_1–42_ +curcumin+MSAB vs. Aβ_1–42_ +curcumin mice. ^$^
*p* < 0.05; ^$$^
*p* < 0.01, Aβ_1–42_ +curcumin+Fc vs. Aβ_1–42_ +curcumin mice. Swimming speeds of different groups of mice (down). (**C**) Percentage of swimming time (%) in the PQ in the MWM test of the control and Aβ_1–42_ mice, df = 14; *t*-test. * *p* < 0.05 vs. control mice. (**D**) Percentage of swimming time (%) in the PQ in the MWM test of Aβ_1–42_, Aβ_1–42_+MSAB, Aβ_1–42_+Fc, Aβ_1–42_ +curcumin, Aβ_1–42_ +curcumin+MSAB, and Aβ_1–42_+curcumin+Fc mice. *n* = 8 mice per group, two-way ANOVA, followed by Tukey’s test. * *p* < 0.05 vs. Aβ_1–42_ vs. Aβ_1–42_ +curcumin mice. ^#^
*p* < 0.05; ^##^
*p* < 0.01 vs. Aβ_1–42_ +curcumin mice. Representative images of swimming paths in the hidden platform test (days 3–7) of different groups on day 6 (**A**,**B**) and in the spatial probe test (day 8: the platform was removed) from different groups, which is representative and can show the differences in search time and tracks (upper panels), blue circles, and the position of the platform. The roman numbers mean the four quadrants in the maze.

## Data Availability

Raw data supporting the conclusions of this paper will be provided without reservation by authors.

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
