# Peer review of "Curcumin Improves Neurogenesis in Alzheimer’s Disease Mice via the Upregulation of Wnt/β-Catenin and BDNF"

_ijms, 2024, doi:10.3390/ijms25105123_

Round 1

Reviewer 1 Report

Comments and Suggestions for Authors

Dear Editor

The article “Curcumin improves neurogenesis of Alzheimer’s disease mice 2 via upregulating Wnt/β-catenin and BDNF” has a good subject. Before proceeding, it needs major revision.

1- The study needs to check for grammar errors.

2. Please clarify the introduction; Explain the relationship between Wnt/β-catenin, BDNF, and curcumin in more detail. It needs to be expanded, especially by supporting it with current articles in the literature.

4- Please remove references 21 and 22. The result section is a section where only results should be reported. Therefore, references 21 and 22 should be removed and explained in the discussion section.

5- The material and method should be detailed in an understandable manner. It contains many grammatical errors, and the scope is not fully understood.

6- Please add include and exclusion criteria to the method section.

8-The discussion section was well designed, but the authors' own findings and the literature were not discussed sufficiently and were incomplete in this context. The discussion needs to be expanded with literature information and a comprehensive comparison should be made with its own results.

Comments on the Quality of English Language

The study needs to check for grammar errors.

Reviewer 2 Report

Comments and Suggestions for Authors

This is an interesting study reporting some of the neuroprotective effects of curcumin, a compound widely studied for its health benefits. Here, the authors investigated the potential of curcumin to improve neurogenesis and cognitive function in an animal model of Alzheimer’s disease. Generally, the work is well-conducted, figures are easy to understand and the findings would be a relevant addition to the current knowledge. I have some questions for the authors that should be addressed in the revised version of the manuscript.

1. In the introduction (L. 56-63), the authors comment on some of the biological effects of curcumin that are documented in the literature. However, I believe that this part should better contextualize and present information about what is already known about the neuroprotective effects of curcumin, especially the pathways related to neuroprotection.

2. In general, the methodology needs to be explained in more detail allowing other researchers to reproduce the protocols.

3. L. 353, Section “4.1. Experimental objects”: Please change the word “objects”. Also, the total number of animals used for the study should be reported, and their weight.

4. Provide the statistical analysis and the n in each figure legend (one or two-way analysis of variance, post hoc test).

5. What was the vehicle for Aβ1-42 injection? Please include it in the methods section.

6. In the Discussion (L. 289-294), please provide a reference.

7. Correct some typing and punctuation errors. Ex: “Howerver” (L.318).

Reviewer 3 Report

Comments and Suggestions for Authors

In this study, using an injection of β-amyloid in mice model of Alzheimer's disease, the authors examined the function of curcumin in reversing adult neurogenesis and cognitive impairment. They deepened their research to clarify the potential molecular mechanisms behind this beneficial effect of curcumin. Although the efficacy of this phytomolecule in stimulating neurogenesis and enhancing cognitive abilities is described in the literature, this manuscript provides new insights into the pathways involved in the context of Alzheimer’s disease in vivo. Overall, the manuscript is informative, but there are a substantial number of grammar and writing style mistakes that need to be corrected before it can be considered for publication.

-Title: “Curcumin improves neurogenesis of Alzheimer’s disease mice via upregulating Wnt/β-catenin and BDNF” should be changed to "Curcumin improves neurogenesis in Alzheimer's disease mice via upregulation of Wnt/β-catenin and BDNF". 

- In my opinion, the authors should add a graphical abstract resuming the pathways proposed to be involved in the beneficial effect of curcumin and the outcomes (neurogenesis, enhanced cognitive function in mice). 

- Sometimes GSK3β is written with a capital letter, sometimes in lowercase. Please check the whole manuscript and correct it according to the nomenclature guidelines.

- Line 10: Please change “in AD mice and the potential mechanism” in “in AD mice and its potential mechanism.”.

- Line 28: Change in “and cognitive impairment”.

- Line 69: I believe that the timing after BrdU administration can be omitted in the title, as it is already mentioned in the text.

- Line 83: Change “was” to “were”. 

- Figure 1: Please, describe statistics in the caption, and indicate the symbols (* and #) meaning.

- Figure 1 and Figure 2: Please upload charts with a larger size.

- Figure 1: I would also suggest omitting the indication “14-days” in the y-axis, as it is already mentioned in the text and it overlaps with the x-axis name; furthermore, it is written “Curcucmin”, please correct (this mistake is repeated in other charts). Always in this section, I believe that it would be better to display a single result, showing BrdU+ and DCX+ signals alone, and then a picture in merge. The same criteria can be applied for BrdU+/NeuN+ cells.

- Line 145: Please check the grammar. I suggest reformulating “Moreover, CREB phosphorylation was decreased”.

- Figure 3D: The chart indicates a decrease in BDNF levels in Aβ+Curcumin experimental group, while in the text you state the opposite. Please correct. 

- Figure 3E: Please indicate the experimental group belonging to each line of the WesternBlot.

- Lines 159 to 161: Please check the grammar in this sentence.

- Line 160: As you mentioned α7nAChR, please briefly remark its role in neurogenesis.

- Figure 4: The charts overlap with WesternBlot gel pictures. Please re-order.

- Materials and Methods section: Even if references are provided, I would like the authors to briefly describe the methodologies used, particularly for the 4.7 "Morris Water Maze section". Having all the information in one paper would be much more helpful for the reader than having to open multiple sources.

Comments on the Quality of English Language

The manuscript needs extensive English editing.

Round 2

Reviewer 1 Report

Comments and Suggestions for Authors

Relevant corrections have been made and the scope of the article has been expanded. There are minor language errors. Thank you to the authors.

Comments on the Quality of English Language

There are minor language errors.